# Investigating lytic polysaccharide monooxygenase-assisted wood cell wall degradation with microsensors

Hucheng Chang [1], Neus Gacias Amengual [1], Alexander Botz[1], Lorenz Schwaiger[1], Daniel Kracher[1,2], Stefan Scheiblbrandner[1], Florian Csarman [1] & Roland Ludwig [1] ✉

Lytic polysaccharide monooxygenase (LPMO) supports biomass hydrolysis by increasing saccharification efficiency and rate. Recent studies demonstrate that $H_2O_2$ rather than $O_2$ is the cosubstrate of the LPMO-catalyzed depolymerization of polysaccharides. Some studies have questioned the physiological relevance of the $H_2O_2$-based mechanism for plant cell wall degradation. This study reports the localized and time-resolved determination of LPMO activity on poplar wood cell walls by measuring the $H_2O_2$ concentration in their vicinity with a piezo-controlled $H_2O_2$ microsensor. The investigated *Neurospora crassa* LPMO binds to the inner cell wall layer and consumes enzymatically generated $H_2O_2$. The results point towards a high catalytic efficiency of LPMO at a low $H_2O_2$ concentration that auxiliary oxidoreductases in fungal secretomes can easily generate. Measurements with a glucose microbiosensor additionally demonstrate that LPMO promotes cellobiohydrolase activity on wood cell walls and plays a synergistic role in the fungal extracellular catabolism and in industrial biomass degradation.

The discovery of lytic polysaccharide monooxygenase (LPMO) featuring a surface exposed active-site with a type-2 copper center, has changed the perception of fungal lignocellulose degradation[1–4]. LPMOs employ an oxidative rather than a hydrolytic mechanism to cleave glycosidic bonds in cellulose[3,5], hemicelluloses[6], chitin[2], starch[7], xylan[8] and soluble cello-oligosaccharides[9] (carbohydrate-active enzyme CAZy auxiliary activities AA9-12, AA13-17). LPMOs have been widely identified in fungi[8], bacteria[2] and others. The research on LPMOs is driven by their ability to promote biomass saccharification[10–12]. LPMO has also been found to play an opposing role in the pathogen-defense mechanism of plants[13,14]. The elucidation of LPMO's reaction mechanism, its activity on natural substrates and its interaction with other enzymes is a prerequisite for advanced applications in plant biomass hydrolysis like the selective depolymerization in biorefineries and for understanding its role in physiological

processes. Previous work demonstrated that each LPMO, irrespective of its origin, needs an electron donor to activate the copper center to prime the catalytic cycle[15]. The flavocytochrome cellobiose dehydrogenase (CDH) is the proposed native reductase of LPMO[16,17]. Once activated, the copper center utilizes an oxygen species as cosubstrate to hydroxylate the C1 or the C4 atom of the polysaccharide substrate, thereby breaking the glycosidic bond[9,18,19]. The nature of the cosubstrate has initially been assumed to be $O_2$, but research in the last years brought forth ample evidence for $H_2O_2$ as actual cosubstrate[20–25]. The catalytic efficiency of LPMO for polysaccharide depolymerization is much higher when adding $H_2O_2$ compared to using the preliminary reduction of $O_2$ to $H_2O_2$ either directly by the added electron donor (reductant) or via an oxygen uncoupling mechanism of the reduced, non-substrate bound LPMO[20,26]. The reaction rate of the activated LPMO9A from *Trichoderma reesei* (*Tr*LPMO9A) and $H_2O_2$ was three

[1]Department of Food Science and Technology, Institute of Food Technology, University of Natural Resources and Life Sciences, Muthgasse 18, 1190 Vienna, Austria. [2]Present address: Institute of Molecular Biotechnology, Graz University of Technology, Petersgasse 14, 8010 Graz, Austria. ✉e-mail: roland.ludwig@boku.ac.at

orders of magnitude faster than for $O_2$[23]. The reported $k_{cat}/K_M$ values of bacterial and fungal LPMOs for $H_2O_2$ are in the order of $10^6 M^{-1} s^{-1}$ [24,27], which is comparable to catalytic efficiencies of fungal peroxygenases[28] and peroxidases[29]. Conversely, the catalytic efficiency of a fungal LPMO for $O_2$-dependent substrate oxidation was measured to be $10^3 M^{-1} s^{-1}$ [22]. In an industrial setting, saccharification rates and yields for treatment of both pure cellulose and birch samples increased when $H_2O_2$ was supplied for LPMO and cellulases[21]. While aiming for higher efficiency of cellulose saccharification, $H_2O_2$-driven LPMO catalysis has not yet been studied at the structural level of wood cells which form the natural environment of fungal hyphae and their secreted enzymes. To measure the peroxygenase activity of LPMO on plant cell walls under near-natural conditions and in the presence of cellulases is an essential step to understand the enzyme's natural function.

The study of LPMO's reaction kinetics and interaction with other enzymes is hampered by the necessity of an analytical method that overcomes the experimental difficulties associated with the investigation of enzymes bound to heterogeneous and structurally complex biopolymers in plant cell walls[30,31]. In the secondary cell wall, the main target of fungal exoenzymes, the cellulose microfibrils are surrounded by hemicellulose and inter-connected lignin networks, causing the recalcitrance of lignocellulosic biomass towards microbial deconstruction[32]. LPMOs are active on cellulose and hemicelluloses and are a substantial constituent of fungal secretomes[6]. Several studies report that LPMO can either promote or impede cellobiohydrolase activity depending on the studied enzymes and substrates[12,33,34]. A combination of TrCel6A and TrLPMO9A showed a 2–2.5-fold increase in the degradation of amorphous cellulose compared to the same dosage of individual enzyme but showed negligible effect on crystalline cellulose[33]. The addition of C1-oxidizing LPMO from Thielavia terrestris increased the glucose production from bacterial cellulose when administered with TrCel6A or TrCel7B, however, it decreased glucose production when administered with TrCel7A[12]. LPMOs alter the physical accessibility and chemical structure of cellulose chains and can promote the activity of cellobiohydrolases[24,25]. This effect is of great significance since cellobiohydrolases are the most abundant enzymes in the secretomes of most wood-decaying fungi[35,36].

Although LPMO activity can be measured by different assays, no currently available method is able to detect its activity at the microscale when bound to a natural substrate. In this work a $H_2O_2$ microsensor was developed and positioned closely above a poplar

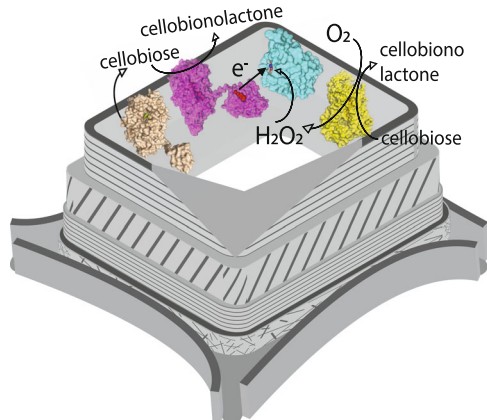

**Fig. 1 | Schematic illustration of the cascade reactions of the bound enzymes on wood cell walls.** In this study, fungal cellobiohydrolases (light brown, model based on PDB IDs: 7cel and 5x34) and ChCDH (yellow, PDB ID: 4qi5) are used to imitate fungal $H_2O_2$ production on poplar wood cell walls. The $H_2O_2$ is consumed by NcLPMO9C (cyan, PDB ID: 4d7u) after priming by its auxiliary electron donor NcCDH (magenta, PDB ID: 4qi7). Oxygen is provided from air.

wood microtome slice by using a scanning electrochemical microscopy (SECM) platform to investigate LPMO's peroxygenase activity on the surface of wood cell walls in a physiologically relevant environment. Since a microsensor is barely limited by mass transfer and only marginally depletes the analyte, the set-up allows the real-time detection of $H_2O_2$ consumption by LPMO bound onto wood cell walls for the first time. Cellobiohydrolases from T. reesei and a CDH variant from Crassicarpon hotsonii (syn. Myriococcum thermophilum) that possesses a 30-fold higher oxygen turnover than the native CDH were used to hydrolyze cellulose in wood cell walls and generate $H_2O_2$ in situ for Neurospora crassa LPMO9C[37] (Fig. 1). The enhanced $H_2O_2$ production rate of the ChCDH variant enabled us to shorten the experimental time ($\leq 4$ h) to maintain the microsensors' performance and reduce positioning errors from substrate swelling. A glucose-detecting microbiosensor was additionally used to study the effect of $H_2O_2$-driven LPMO activity on the activity of cell wall-bound cellobiohydrolases.

## Results

### Positioning of a micro(bio)sensor

An SECM platform is composed of a stepper motor with a piezo module, a lock-in amplifier, a potentiostat, an inverted fluorescence microscope and a positioning microscope and the parts are placed in two adjacent Faraday cages (Fig. 2a and Supplementary Fig. 1). A shear-force based SECM mode was employed to position a micro(bio)sensor in close vicinity to the cell walls in cross-section ultramicrotome slices of poplar wood. The resonance frequency of a fixed microsensors (e.g., ~396 kHz) was determined by searching the magnitude difference of the vibrating tip operation in air and in solution (Fig. 2b).

The microsensor was moved towards the poplar wood slice surface, and the approaching stopped once the change of the shear-force vibration magnitude overcame 5%, which occurred at a distance of ~300 nm above the sample surface (Fig. 2c, d). Finally, the microsensor was retracted to 25 μm above the surface of the poplar wood cells in all experiments (Fig. 2d, e).

### Performance of the $H_2O_2$ microsensor

A platinum wire was embedded in a quartz capillary by laser-assisted electrode pulling and resulted in ultramicroelectrodes with a tip length of 1.0–1.5 cm, an electrode diameter between 1–2 μm, and an outer diameter of $20 \pm 5$ μm (Fig. 2f, g). With an electrodeposition method, the Pt microelectrode was modified by a Prussian blue layer (Fig. 3a). The $H_2O_2$ microsensors were characterized by a pair of redox waves (Supplementary Fig. 2) consistent with the redox cycle between Prussian blue and Prussian white[38,39]. The scanning electron microscope images showed the Pt ultramicroelectrode is covered with Prussian blue particles (Fig. 3b, c). Cyclic voltammograms show that $H_2O_2$ reduction reaction on a Prussian blue-modified Pt ultramicroelectrode starts from 0.1 V vs. Ag|AgCl during the cathodic scan (Supplementary Fig. 3). The Prussian blue-modified $H_2O_2$ microsensor shows a negligible current for oxygen reduction reaction compared to a bare Pt microelectrode which is a necessary feature to perform measurements under natural, air-saturated conditions (Supplementary Figs. 4 and 5). The high selectivity of the electrocatalyst for $H_2O_2$ reduction is consistent with previous reports[40] and ensures an extremely low background signal allowing the selective and sensitive detection of $H_2O_2$ in presence of $O_2$. The amperometric response of reducing various concentrations of $H_2O_2$ stabilizes after 40 s (Fig. 3d). Steady-state currents of $H_2O_2$ reduction were calculated as mean values from the time interval between 40 and 50 s and used to plot a calibration curve (Fig. 3e). A linear correlation was found in the measured range between 25–200 μM $H_2O_2$ and a sensitivity of 0.078 pA μM$^{-1}$ μm$^{-2}$ was calculated. The sensitivity of the shown $H_2O_2$ microsensor was re-examined after four hours of a typical experiment to be 0.066 pA μM$^{-1}$ μm$^{-2}$. This 14.3% reduction of the sensitivity is acceptable but meanwhile shows the

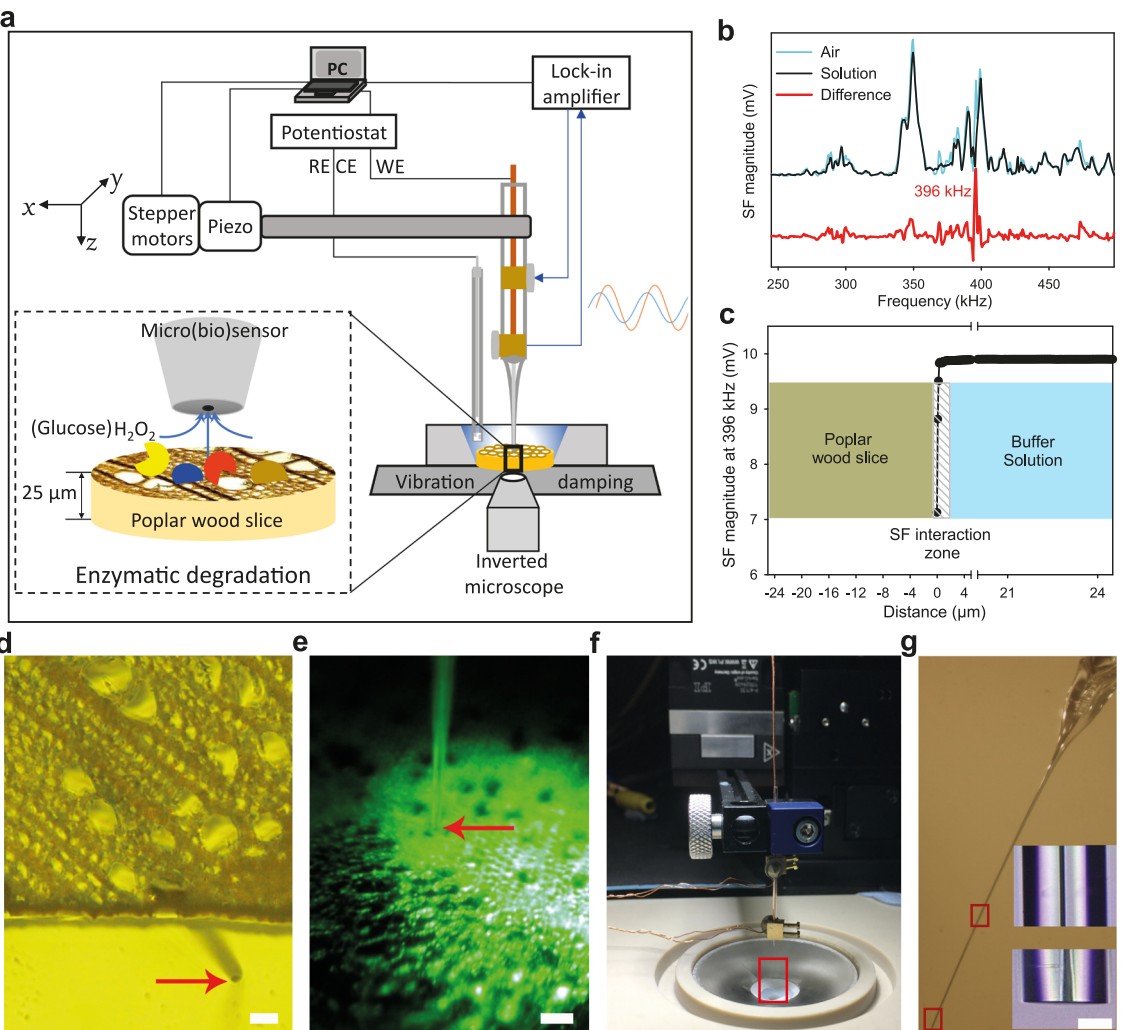

**Fig. 2 | Experimental set-up to study the enzymatic activity on poplar wood cell walls. a** Schematic representation of the instruments combined in the SECM platform. **b** Frequency spectra were acquired using the shear-force mode and two piezo-crystals to oscillate the microsensor and measure its attenuation in air (blue) and in solution (black). The red line represents the magnitude difference which indicates 396 kHz as the preferable resonance frequency of the used microsensor. The resonance frequencies of all used micro(bio)sensors were in the range of 340–416 kHz. **c** The approach curve of a microsensor shows the magnitude changes at the resonance frequency with decreasing distance from the poplar wood surface. **d** Image of an approached microsensor laterally near a poplar wood slice (in buffer solution) obtained using the inverted microscope. **e** Image of a positioned microsensor above a poplar wood slice (in buffer solution) from the positioning video microscope. **f** A microsensor with the mounted piezo plates is fixed on the black cantilever of the positioning unit. The view in red rectangle is zoomed in and shown in **g**. **g** Light microscopic images of a Pt ultramicroelectrode. High-magnification images of a middle and a tip segment were shown in the bottom right corner. Scale bars, 50 μm (**d**); 50 μm (**e**); 10 μm (**g**).

time limit of highly sensitive, real-time measurements of $H_2O_2$ microsensors.

## Localized formation and detection of $H_2O_2$

A cell wall-associated enzymatic production of $H_2O_2$ mimics the fungal degradation mechanism and provides a more realistic scenario than external cosubstrate addition when studying LPMO activity. The combination of purified *T. reesei* cellobiohydrolases and a *C. hotsonii* CDH variant capable of binding to poplar wood slices by their carbohydrate-binding modules were used to produce $H_2O_2$ in a catalytic cascade reaction. The cellobiohydrolases activity converts cellulose into cellobiose which is oxidized by CDH using $O_2$ as an electron acceptor. The *C. hotsonii* CDH variant with a 30-fold higher oxygen turnover than the wild type[37,41] was used to obtain $H_2O_2$ concentrations that can be reliably quantified by a microsensor within the experimental time limits. Upon adding both enzymes, the localized $H_2O_2$ concentration in the close vicinity of the poplar wood cell walls increased over 120 min and reached $106.1 \pm 6.7$ μM at the end (Fig. 4a).

The observed, almost linear increase demonstrates a relatively stable enzymatic activity during the experiment. No $H_2O_2$ production was detected in the control experiment without *Ch*CDH. An experiment in which *Ch*CDH was substituted by glucose oxidase (GOX) and β-glucosidase resulted in an almost two-fold higher $H_2O_2$ concentration of $208.5 \pm 6.2$ μM, which demonstrates that the availability of the substrates cellulose and cellobiose were not rate-limiting for the CDH catalyzed reaction. These results verify that the microsensor is able to conduct real-time measurements of $H_2O_2$ in the close vicinity of poplar wood cell walls.

## Detection of local $H_2O_2$ consumption by LPMO

The developed microsensor approach is currently the only available method to measure the peroxygenase activity of LPMO on an intact plant cell wall sample. The recently published turbidimetric[42] and photometric assays[24] depend on purified and disperse substrates. To study LPMO activity on microtome slices of poplar wood as a realistic approximation of the natural conditions, *N. crassa* LPMO9C and

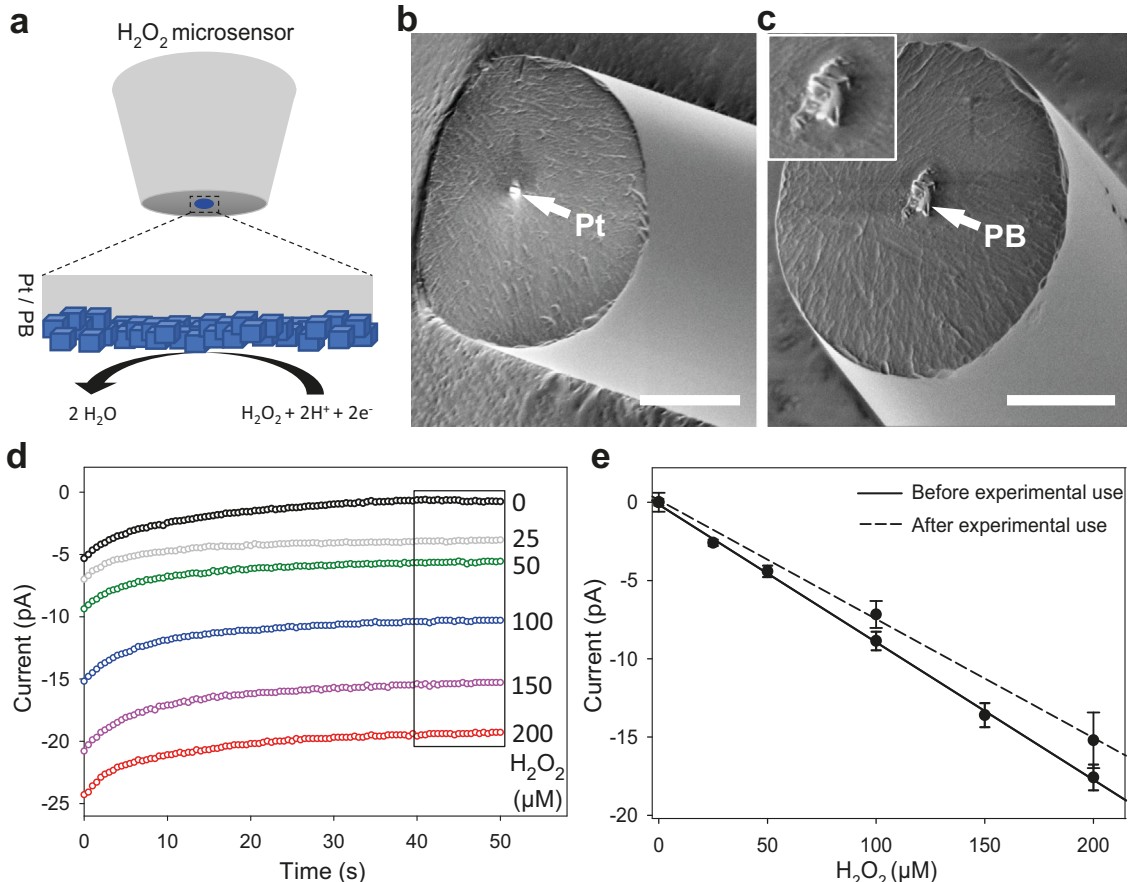

**Fig. 3 | Characterization of H₂O₂ microsensors. a** Schematic principle of Prussian blue based $H_2O_2$ microsensor. **b**, **c** are scanning electron microscope images of a Pt ultramicroelectrode and the Prussian blue modified Pt ultramicroelectrode. Scale bars, 10 μm (**b**); 10 μm (**c**). **d** Amperometric response of a $H_2O_2$ microsensor to 0 μM (black,), 25 μM (gray), 50 μM (green), 100 μM (blue), 150 μM (magenta) and 200 μM (red) $H_2O_2$. **e** Calibration plots of the $H_2O_2$ microsensors before (solid line) and after 4-h experimental use (dashed line). The currents were extracted from the mean values between 40 s and 50 s of the measurements. Data in **e** are shown as means values, and error bars show SD ($n = 3$, independent experiments).

$Nc$CDHIIA were delivered to the poplar wood slice surface using a microsyringe with the aid of a video microscope. *N. crassa* CDH (not the $H_2O_2$-producing *C. hotsonii* CDH variant) was used as an electron donor because it is the natural electron donor for *N. crassa* LPMOs[15,43]. It also has a carbohydrate-binding module that ensures its binding to the cell walls. The $Nc$CDH was injected 3 min after LPMO9C to allow the LPMO binding to the cell walls. Before adding LPMO and $Nc$CDH, the $H_2O_2$ production rate was similar to the reaction rate in the reference experiment (Fig. 4a) with 0.85 vs. 0.88 μM min⁻¹, respectively. Immediately after the addition, the $H_2O_2$ concentration decreased quickly and then leveled off after ~30 min; however, the $O_2$ concentration in the experiment was only reduced by the action of *C. hotsonii* CDH, but not by the addition of *N. crassa* LPMO or CDH (Fig. 4b). This result that LPMO does not consume any measurable amount of $O_2$ has been observed in at least three independent experiments and shows the preference of LPMO for $H_2O_2$ in the presence of $O_2$.

The stop of observable $H_2O_2$ consumption can have three different reasons: (i) reaching an equilibrium between $H_2O_2$ formation and consumption, (ii) a slowdown of LPMO activity due to substrate depletion, and (iii) deactivation of LPMO because of the high $H_2O_2$ concentration. The injection of three different LPMO concentrations (Fig. 4c) showed that it is not a depletion of substrate available to LPMO but most likely a combination of deactivation and reaching a kinetic equilibrium. This is evident from the increase in $H_2O_2$ concentration from 210 to 240 min after adding 0.5 μM LPMO and the stagnation in this time when 3 μM LPMO was added. The high accumulated $H_2O_2$ concentration was necessary for the detection by the

developed microsensors. In nature, such high $H_2O_2$ concentrations are unlikely because of the lower production rate and the continuous consumption by peroxygenases, peroxidases and catalases[44]. Previous studies have found that the oxidation of uncoupled LPMOs with $H_2O_2$ inactivates the enzyme quickly[20,45]. The structural complexity of wood cell walls resulting in lower accessibility of binding sites for LPMO could intensify their oxidative deactivation. The activity of *N. crassa* LPMO9C on poplar wood with $H_2O_2$ can be estimated from the initial $H_2O_2$ consumption rates, which were 3.2–16.7 μM min⁻¹ depending on the added LPMO concentration (0.5–3 μM). The calculation of a specific activity or turnover number is difficult, since from the added LPMO molecules, only a fraction was bound to its substrate and therefore active. The lowest estimate for a turnover number when considering all added LPMO molecules active was 0.1 s⁻¹ calculated from the data presented in Fig. 4c. These values are still six times higher than the reported turnover numbers on amorphous cellulose with $O_2$ as a cosubstrate[22,46].

**Excessive $H_2O_2$ deactivates LPMO**

The previous experiments have shown the LPMO can consume the accumulated $H_2O_2$ in the surroundings of wood cell walls at a high rate. An interesting question is the actual $H_2O_2$ concentration when the LPMO is present at the very beginning of the experiment. Will $H_2O_2$ accumulate, or will LPMO consume it and keep it at a low concentration that is not harmful to the enzymes and possibly the fungus?

To answer this question, 1 μM *N. crassa* LPMO9C and 0.5 μM CDHIIA were simultaneously applied with the $H_2O_2$ producing

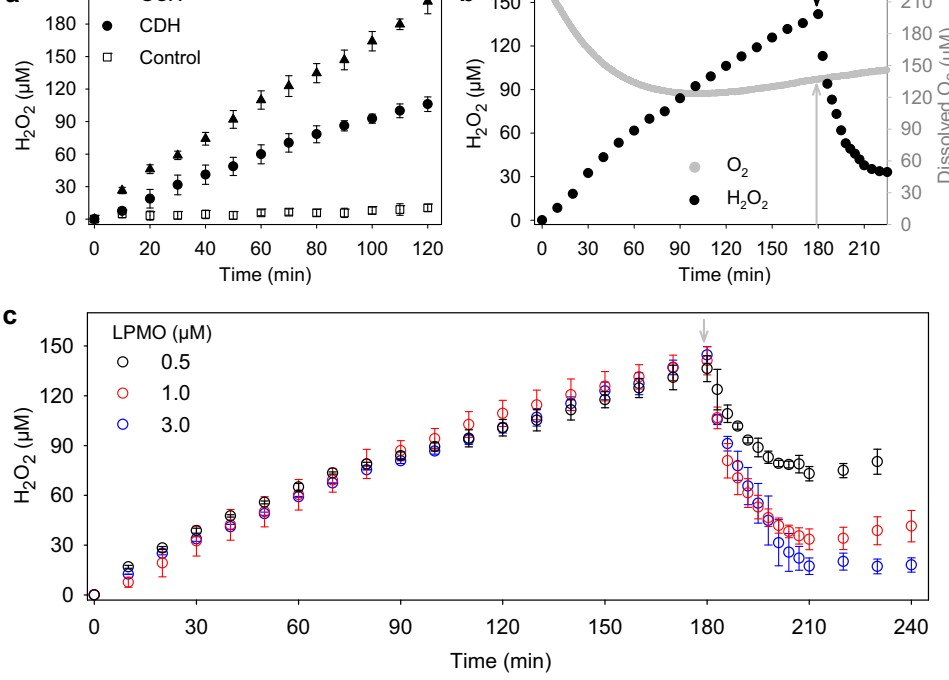

**Fig. 4 | Detection of local $H_2O_2$ formation and consumption in the vicinity of wood cell walls. a** The graph shows time courses of $H_2O_2$ formation by 2 mg mL$^{-1}$ cellobiohydrolases and 1 μM *C. hotsonii* CDH variant (filled circles) or 2 mg mL$^{-1}$ cellobiohydrolases and 1 mg mL$^{-1}$ β-glucosidase in combination with 1 μM GOX (filled triangles) during hydrolysis of poplar wood cell walls in the absence of LPMO. In the control experiment (empty squares), only 2 mg mL$^{-1}$ cellobiohydrolase has been applied. **b** The change of $O_2$ (gray circles) and $H_2O_2$ concentration (black circle) in the vicinity of wood cell walls before and after LPMO catalysis. The additions of 1 μM LPMO9C and 1.0 μM *N. crassa* CDHIIA are indicated by a black

arrow. **c** Time courses of $H_2O_2$ consumption by 0.5 μM (black circle), 1.0 μM (red circle), and 3.0 μM (blue circle) LPMO9C in combination with 1.0 μM *N. crassa* CDH. Their additions after 180 min are indicated by a gray arrow. In the experiments shown in **b**, **c**, 2 mg mL$^{-1}$ cellobiohydrolases and 1 μM *C. hotsonii* CDH variant are applied since the beginning to produce $H_2O_2$. All experiments were conducted in a 50 mM sodium acetate buffer, pH 5.5 at ~20 °C. The reference experiment without addition of LPMO is given in Fig. 5a (black circles). Data in **a** are shown as means values, and error bars show SD ($n = 3$, independent experiments). Data in **c** are shown as means values, and error bars show SD ($n = 2$, independent experiments).

enzymes. In the initial 50 min of the reaction, $H_2O_2$ was not detected in the vicinity of the wood cell walls. In the following 70 min (50–120 min), trace amounts of $H_2O_2$ were detected (Fig. 5a). These concentrations were lower than the limit of quantitation of the used microsensors (~10 μM), making them less accurate. However, a slow accumulation of $H_2O_2$ was indeed observed, which could be the reason for increasing LPMO deactivation. After 120 min, the $H_2O_2$ concentration increased at an approximately equal rate as the control reaction without LPMO. At this point, we assumed that nearly all the LPMO enzymes were deactivated and ceased using $H_2O_2$. Generally, during the first 120 min, remarkably little $H_2O_2$ was detected in the surroundings of the poplar wood cell walls. The presence of LPMO9C therefore efficiently suppressed $H_2O_2$ accumulation by using it for catalysis. Based on the total produced $H_2O_2$, the total turnover number of LPMO during the first 120 min was calculated to be $103 ± 2$. The results also indicated that LPMO can help maintain a very low $H_2O_2$ concentration under physiological conditions, which cannot be easily detected and will significantly reduce enzyme deactivation of hyphal damage. The rapid inactivation of LPMOs in the presence of a high $H_2O_2$ concentration has been reported in previous studies[45]. We propose that under natural conditions and even lower physiological $H_2O_2$ concentrations produced by fungi the total turnover number of LPMO will be much higher than previously suggested[22]. To verify the activity of cellobiohydrolases and LPMO, a glucose microbiosensor was employed to quantify the released cellobiose and cello-oligosaccharides in both reactions (Fig. 5b). In both experiments, the activity of the cellobiohydrolases continuously increased the glucose concentration, but the presence of LPMO boosted the amount of reaction product by 19.6% after 120 min. The synergy of the enzymes became less obvious at longer incubation periods which coincided with the deactivation of LPMO after 120 min.

## Localization of LPMO and CDH on cell walls

To investigate their binding onto the wood cell walls during $H_2O_2$-driven catalysis, LPMO and *Nc*CDH were tagged with different fluorescent moieties to check their selective localization on wood cell walls. Fig. 6a, b showed that both enzymes were mainly bound to the S3 secondary cell walls that are rich in cellulose. The unspecific binding of *Nc*CDH was found on the middle lamella and cell corners regions (Fig. 6a). The brightfield image showed the solid cell walls (Fig. 6c). The overlay of both fluorescent channels and the lignin autofluorescence detected between 415 and 550 nm showed abundant overlaps of enzyme binding sites on the cell walls (Fig. 6d). This implied a frequent interaction between these two enzymes during catalysis, especially when LPMO was reduced by *Nc*CDH, which increased the binding affinity of the LPMO[47] (Supplementary Fig. 6).

## LPMO promotes cellobiohydrolase activity

For this experiment, poplar wood sections were pretreated with LPMO9C to modify the cell walls before adding *T. reesei* cellobiohydrolase acting from the reducing-end (Cel7A) or the nonreducing-end (Cel6A) of a cellulose chain. By adding β-glucosidase, glucose was obtained as the final hydrolysis product, and a glucose microbiosensor was used to assess the cellobiohydrolase activity on poplar wood cell walls. The microbiosensors were calibrated by fitting the steady-state currents vs. the glucose concentrations (Supplementary Fig. 7). The collected data was considered only if the glucose microbiosensors retained a final sensitivity >75% of the initial value after completion of the measurement (Supplementary Fig. 8). The glucose production measured on the LPMO-pretreated poplar wood slices after 30 min increased by 40 and 44% for *Tr*Cel6A and *Tr*Cel7A, respectively, compared to the untreated samples (Fig. 7). The increased turnover of

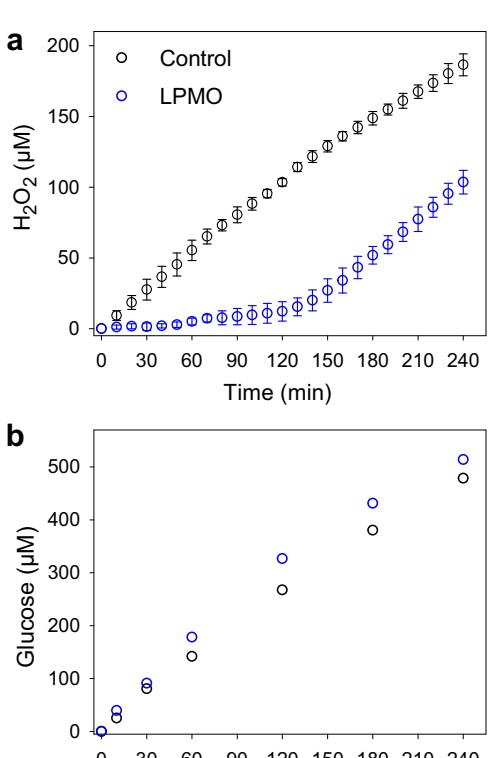

**Fig. 5 | LPMO fully uses the produced $H_2O_2$ in poplar wood cell walls. a** Time courses of $H_2O_2$ formation on wood cell walls in the presence (blue circles) and absence (black circles) of *Nc*LPMO9C while loading 10 mg mL$^{-1}$ cellobiohydrolases and 1 μM *Ch*CDH in 50 mM sodium acetate buffer, pH 5.5, at 20 °C. **b** Time courses of glucose formation on wood cell walls in the presence (blue circles) and absence (black circles) of *Nc*LPMO9C while loading 10 mg mL$^{-1}$ cellobiohydrolases and 2 mg mL$^{-1}$ β-glucosidase in 50 mM potassium phosphate buffer, pH 6.0, at 20 °C. 1 μM *Nc*LPMO9C with 0.5 μM *Nc*CDHIIA are used in both experiments. Data in **a** are shown as mean values, and error bars show SD (*n* = 3, independent experiments), in **b** data of single control measurements are presented.

both cellobiohydrolases after LPMO pretreatment can be either attributed to the formation of additional free ends on cellulose, acting as starting sites for cellobiohydrolases, or the removal of hemicelluloses, which may improve the accessibility of cellulose fibrils.

## Discussion

Ultramicroelectrodes used as micro(bio)sensors have been used previously in SECM to measure the release or consumption of neurotransmitters, $H_2O_2$, or glucose from live cells[48] or bacteria[49]. Enzymatic degradation of woody biomass essentially occurs at the wood cell walls. We used the SECM set-up for the positioning of the micro(bio) sensors in a fixed, close position to the wood cell wall to monitor temporal changes in the $H_2O_2$ or glucose concentration while loading with various oxidoreductases under experimental conditions closely resembling the natural environment of fungal biomass breakdown.

A Prussian blue thin film endowed the $H_2O_2$ microsensor high sensitivity and selectivity when operating in a complex reaction mixture containing several enzymes and other compounds. The results from the control reaction in Fig. 4a verified that none of the matrix components interfered with the measured signals and the measured currents accurately reflect the $H_2O_2$ concentration during enzymatic degradation of wood cell walls. Additionally, the microsensor methodology avoids the interfering side reactions from the reductants and oxidants used for other LPMO activity assay methods[24,50].

In nature, $H_2O_2$ is used by both brown-rot and white-rot fungi to degrade woody biomass through enzymatic and nonenzymatic

reactions. Oxidases and also dehydrogenases from the glucose–methanol–choline (GMC) oxidoreductase superfamily have been identified as sources of $H_2O_2$, e.g., not only alcohol oxidase from *Gloeophyllum trabeum*[51] or aryl-alcohol oxidase from *Pleurotus eryngii*[52] but also $H_2O_2$-producing copper enzymes such as glyoxal oxidase from basidiomycetes[53]. In this study, an enzyme cocktail containing glycoside hydrolases from *T. ressei* and a *C. hotsonii* CDH that can simultaneously degrade wood cell walls and produce $H_2O_2$ was applied to mimic the complicated $H_2O_2$-producing processes during woody biomass degradation by fungi in nature. Our results clearly showed that *N. crassa* LPMO9C binds to poplar wood cell walls and consumes $H_2O_2$. Although the tip diameter (ca. 1.5 mm) of the used oxygen sensor was much larger than that of the $H_2O_2$ microsensor, the close distance from the poplar wood slice surface ensured a fast response. Since no change in the $O_2$ concentration after LPMO addition was observed, this supports the conclusion that LPMO uses $H_2O_2$ as cosubstrate even in the presence of $O_2$[20]. The time-resolved measurement of $H_2O_2$ and $O_2$ concentrations during LPMO catalysis revealed that LPMO uses $H_2O_2$ as cosubstrate and does not consume the present $O_2$ when bound to wood cell walls.

The $H_2O_2$-driven LPMO catalysis was observed for more than 2 h in a near-natural environment, but eventually, the high amount of $H_2O_2$ produced by the auxiliary enzymes in the experimental set-up leads to LPMO inactivation. The $H_2O_2$-producing enzymes are more stable towards their product than LPMO and are active even after accumulating $H_2O_2$ for up to 4 h in the experiments. The production of $H_2O_2$ by CDH under natural conditions is regulated by many factors, such as the availability of phenol derived electron acceptors (which reduce the oxygen turnover), the pH which has a strong effect not only on the catalytic activity but also the electron transfer to LPMO via CDH's cytochrome domain[54]. The presence of organic acids is an important factor that can therefore not only modulate the production and stability of $H_2O_2$ but has also shown to have a modulating effect on LPMO activity[55].

Our results demonstrate that LPMOs secreted by wood-degrading fungi scavenge $H_2O_2$ present in their surroundings. Although we observed the deactivation of LPMOs in the presence of $H_2O_2$, the peroxygenase activity was maintained for up to 120 min in a set-up with a high concentration of $H_2O_2$-generating enzymes. When fungi degrade woody biomass, different extracellular peroxidases (e.g., manganese peroxidases; catalytic efficiency ($k_{cat}/K_{M\ H2O2}$: $10^6$–$10^7$ M$^{-1}$ s$^{-1}$) maintain the $H_2O_2$ concentration at a relatively low level in the surroundings of wood cell walls[56,57]. Under these conditions, LPMO's peroxygenase activity should be sustained much longer than observed in our experiments.

In this experimental set-up, the precise amount of enzyme and substrate concentration in the interfacial area (infinite diffusion hemisphere of the microsensor could not be quantified). Therefore, kinetic constants of LPMO were only reported as estimates. The minimum estimate of the LPMO turnover number for its polymeric substrate (0.1 s$^{-1}$) is much lower than the $k_{cat}$ for the oxidation of soluble cellopentaose by *N. crassa* LPMO9C (124 s$^{-1}$)[58] and or for the oxidation of insoluble chitin by a bacterial LPMO (5.6 s$^{-1}$)[59]. It is worth noting that the enzyme concentrations in our calculation are grossly overestimated because we considered the total added enzyme instead of the bound enzymes on the wood cell walls. Nevertheless, the $H_2O_2$ microsensor is excellent in measuring the peroxygenase activity of various LPMOs on the surface of different solid substrates, as well as other $H_2O_2$-consuming or $H_2O_2$-producing fungal enzymes such as peroxidases.

Enzyme based microbiosensors enable the determination of non-electroactive compounds. Glucose microbiosensors have been used to measure glucose consumption of *Streptococcus mutans* biofilms[60], and measure the dynamic concentration of endogenous glucose in mammalian brain cells[61]. Here, we adopted an

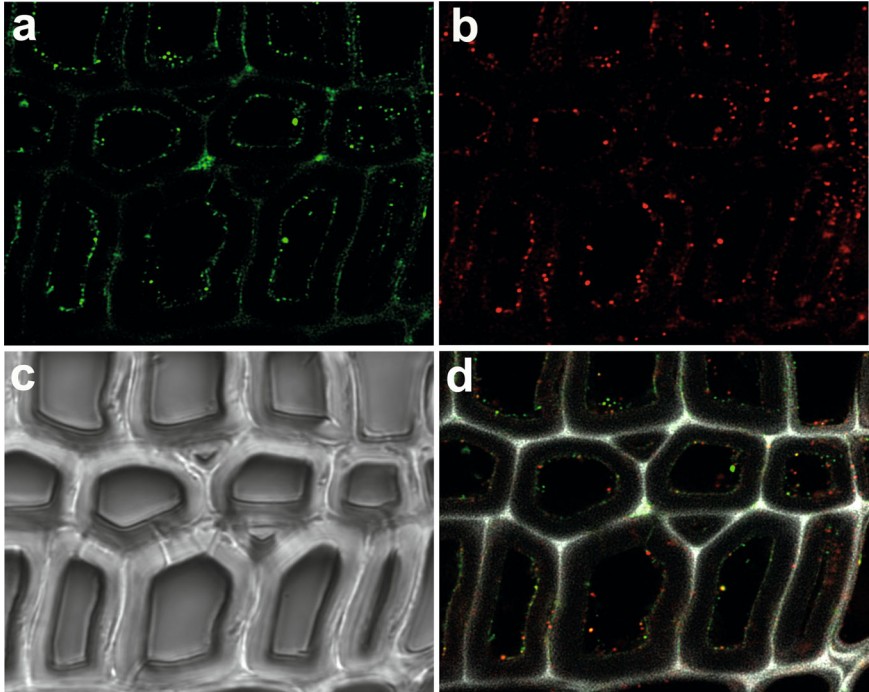

**Fig. 6 | Localization of LPMO and CDH on wood cell walls.** Fluorescence images show the binding of *N. crassa* CDHIIA and LPMO9C onto poplar wood cell walls. **a** Fluorescence from DyLight D550-labeled CDH, and **b** fluorescence from DyLight D633-labeled LPMO. **c** Image from brightfield microscopy shows the wood cell walls. **d** An overlay of both fluorescence channels shows the colocalization of CDH and LPMO. Picture size = measurement area = 70 × 70 μm.

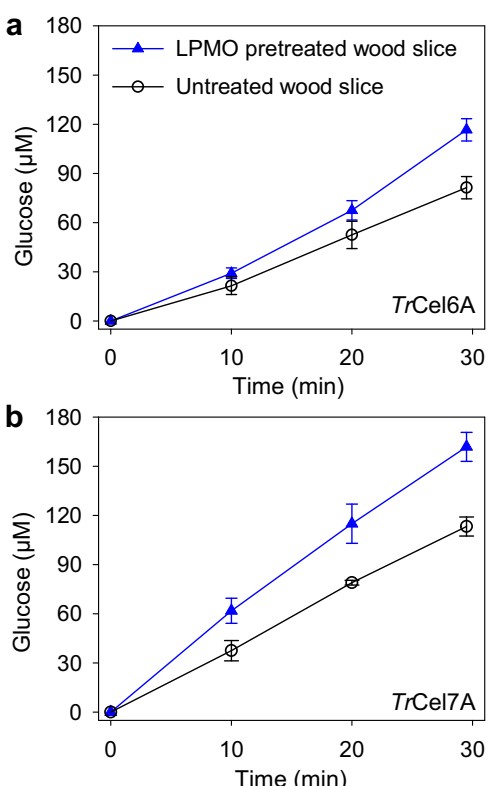

**Fig. 7 | Effect of H$_2$O$_2$-driven LPMOs catalysis on cellobiohydrolases activity.** Glucose production by *Tr*Cel6A (**a**) and *Tr*Cel7A (**b**) while degrading the untreated (black circles) and pretreated (blue triangles) poplar wood slices by *Nc*LPMO9C with its reductant and H$_2$O$_2$ as cosubstrate. All reactions are conducted in a 50 mM potassium phosphate buffer, pH 6.0 at 20 °C. Data are shown as mean values and error bars show SD (*n* = 3, independent experiments).

electrodeposition method to immobilize GOX on a platinum ultramicroelectrode, and the outmost polyurethane coatings effectively protected the immobilized GOX from adsorbing cellulolytic enzymes during degradation experiments, resulting in an improved stability.

LPMOs have been known to act synergistically with glycoside hydrolases during the deconstruction of cellulosic biomass, and supplemented into cellulolytic cocktails in industrial settings of biofuel. The synergy between LPMOs and endoglucanases has been verified using various substrates in different conditions, however, the effect of LPMOs on cellobiohydrolases (exoglucanases) has been disputed by the significant impeding effect of C1-oxidizing LPMOs on reducing-end cellobiohydrolase in recent studies[33,34]. A mechanistic study found that C1-oxidized cellulose chain ends are poor attack sites for reducing-end cellobiohydrolase (Cel7A) but are preferred by nonreducing-end cellobiohydrolase (Cel6A)[34]. Our results showed that both *Tr*Cel7A and *Tr*Cel6A had a higher activity on the wood cell walls oxidized by a C4-oxidizing LPMO9C. Unlike purified cellulosic substrates, the modification of the wood cell wall microstructure is more complex. Since LPMO9C can act on both cellulose chains and hemicellulose networks the increased cellobiohydrolase activity could not only arise from additional ends in the cellulose chain, but also the degradation of the hemicellulose network can increase the accessibility of cellulose chains. This supports that the structural complexity of cell walls affect cellobiohydrolase activity. We found a stronger synergistic activity for the combination of LPMO9C and *Tr*Cel7A, which might originate from the formed C4-oxidized cellulose chain ends that are preferred by this cellobiohydrolase starting from the reducing end. Although a systematic investigation of the effect of C4-oxidizing LPMOs on reducing-end cellobiohydrolase activity has not been reported, a promoting effect of C1-C4 oxidizing LPMOs on reducing-end acting Cel7A has been observed[33]. The effect of LPMO9C in combination with *Tr*Cel6A was less pronounced, but agrees with previous studies reporting LPMOs promoting also nonreducing-end cellobiohydrolase

activity[12,33]. For this study on a natural substrate, we have to consider the effect of hemicellulose degradation by LPMO to be beneficial for both cellobiohydrolases. The increased activity of cellobiohydrolases on LPMO-oxidized wood cell walls adds evidence to the claim that LPMOs act in concert with hydrolytic cellulases in the depolymerization of plant biomass.

## Methods

### Chemicals and instruments

All chemicals were at least of analytical grade and used without further purification. Aqueous solutions were prepared with deionized water (resistivity ca. 18 MΩ cm$^{-1}$ at 25 °C). The surface microstructure of the polished Pt ultramicroelectrodes and Prussian blue modified Pt ultramicroelectrodes were examined using a scanning electron microscope (FEI Apreo). For the electrochemical experiments, a two-electrode set-up was used with a custom-made miniaturized Ag|AgCl (3 M KCl) as both reference electrode and counter electrode. A micro(bio)sensor served as the working electrode. The miniaturized Ag|AgCl electrode was prepared according to the protocol in our previous study, and was benchmarked to a standard Ag|AgCl reference electrode (Bioanalytical Systems, Inc.). In all the amperometric measurements, data was collected at a rate of 2 s$^{-1}$ and corrected for background currents of the used micro(bio)sensors.

The SECM system is comprised of a micromanipulator controller (Sutter Instrument) with a piezoelectric positioning module, a potentiostat (Gamry Reference 600+), a lock-in amplifier, and the software (Sensolytics GmbH) to control all parts. The micromanipulator is placed on a stainless-steel board containing a central sample holder inside a custom-made jacketed double Faraday cage, which is mounted on a concrete-filled steel table supported by four active pneumatic dampers (Supplementary Fig. 1a). A digital single-lens reflex camera is connected with the inverted microscope and coupled to the corresponding software for viewing microstructures of the poplar wood slices.

### Enzymes

LPMO9C and CDHIIA from *N. crassa* were recombinantly produced in *Pichia pastoris* X33 as previously described and purified by a two-step chromatographic procedure employing hydrophobic interaction chromatography (PHE-Sepharose FF) and anion exchange chromatography (DEAE Sepharose FF)[62,63]. The engineering, production and purification of *Ch*CDH were described in a previous study[37]. β-glucosidase from *Aspergillus niger* (Sigma 49291) and glucose oxidase from *A. niger* (Sigma G7141) were used as purchased. The cellobiohydrolases, β-glucosidase and GOX were dosed on a weight basis. *Tr*Cel6A and *Tr*Cel7A were purified from the commercial cellulase cocktail (Sigma C2730) following a published procedure. In brief, the solution was desalted using a Sephadex G25-fine column and cellobiohydrolases were purified by anion exchange chromatography (DEAE Sepharose FF). The *Tr*Cel7A containing fraction was further purified by size exclusion chromatography (Sephacryl S-200 HR). The *Tr*Cel6A containing fraction was subjected to hydrophobic interaction chromatography (PHE Sepharose FF) and finally purified by size exclusion chromatography (Sephacryl S-200 HR). Both cellobiohydrolases were stored in sodium acetate (10 mM, pH 4.8) until further use.

### Poplar wood slices

Poplar wood samples were obtained from white poplar (*Populus alba*) harvested in the winter season (December–February in Vienna, Austria). Small blocks (radial ~5 mm, tangential ~10 mm) were cut out from the branches, debarked and used to prepare 25-μm-thick transverse slices by using a sliding microtome (Microm) with stainless steel blades (Micros). The poplar wood slices were soaked in deionized water for at least 12 h prior to use.

### Pt ultramicroelectrodes

Pt ultramicroelectrodes were prepared using a CO$_2$-laser puller (P-2000, Sutter Instrument) following a previously published procedure[64]. A Pt wire (Ø: 25 μm; Goodfellow) was threaded through and laid at the center of a quartz glass capillary (Ø$_{out}$: 1.0 mm, Ø$_{in}$: 0.5 mm, *L*: 100 mm), which was fixed in the V-groove of the puller bar in the laser puller. The Pt wire was sealed in quartz capillary by heating (parameters: Heat: 800; Filament: 5; Velocity: 128; Delay: 130; Pull: 0) for 6 cycles (20 s laser on and 40 s laser off). Right afterwards the final pulling step was carried out using typical parameters: (Heat: 510; Filament: 2; Velocity: 85; Delay: 128; Pull: 100). The pulled capillaries with an inlaid Pt wire that crosses the whole microneedle tip were obtained. The unpulled backside of Pt wire was soldered to a copper wire using short soldering tin pieces. A custom-made polishing machine that rotates sandpaper and the electrode individually was used to polish the Pt ultramicroelectrodes' tip obtaining a disk geometry. The active electrode area was subsequently characterized in 5 mM [Ru(NH$_3$)$_6$]Cl$_3$ containing 100 mM KCl solution. The electrodes with a well-defined amperometric response (plateau in diffusion limited potential region) were used for further fabrication of micro(bio)sensors.

### H$_2$O$_2$ microsensor

Prussian blue films were deposited on Pt ultramicroelectrodes by cyclic voltammetry running 10–12 scans from 0.4 to 0.75 V vs. Ag|AgCl with a scan rate of 20 mV s$^{-1}$ in a solution containing 4 mM FeCl$_3$, 4 mM K$_3$[Fe(CN)$_6$], 0.1 M KCl and 0.1 M HCl. Afterwards, the Prussian blue modified electrodes were rinsed with distilled water and activated by another 20 cycles between −0.05 and +0.35 V in 0.1 M HCl solution with 0.1 M KCl at a scan rate of 50 mV s$^{-1}$. Their amperometric response at an applied potential of 0.0 V vs. Ag|AgCl was examined in 50 mM sodium acetate buffer solution, pH 5.5 containing varying concentration of H$_2$O$_2$. The Prussian blue modified Pt ultramicroelectrodes with a sensitivity above 60 pA mM$^{-1}$ were qualified for studying enzymatic degradation of wood cell walls and they were also termed as H$_2$O$_2$ microsensors in this article.

### Glucose microbiosensor

Glucose microbiosensors were prepared by immobilization of GOX on the Pt ultramicroelectrodes. GOX was encapsulated within a chitosan polymer matrix by a pH shift-induced deposition of chitosan films[65,66]. In short, chitosan was dissolved in 0.1 M acetic acid and clarified by centrifugation to remove undissolved solid residues. The final chitosan solution with a concentration of ca. 5 mg mL$^{-1}$ was adjusted to pH 4.0 by titration with 1 M NaOH solution. A Pt ultramicroelectrode was thread into a Pt wire coil counter electrode in a solution containing 2.5 mg mL$^{-1}$ chitosan, 15 mg mL$^{-1}$ GOX and 0.5% glutaraldehyde. A cathodic voltage of −2.5 V was applied using a DC power supply for 60–80 s to precipitate chitosan at the ultramicroelectrode surface and encapsulate enzymes within chitosan films simultaneously. The GOX-modified electrodes were further covered by polyurethane coatings according to a published procedure[67].

### Constant-distance electrochemical measurements

Positioning of a micro(bio)sensor in close vicinity to wood cell walls was performed by means of shear-force distance control. Two piezo ceramic plates (Piezomechanik, Gmbh) were mounted in close vicinity to the tip end of the quartz glass tube of a micro(bio)sensor through two brass holders (Supplementary Fig. 1d), at an angle of ca. 45° and 1.0–1.5 cm distance from each other. The micro(bio)sensor was mounted to the motorized micromanipulator controller (Fig. 2f). The top piezoelectric plate, driven by an arbitrary waveform generator, induced a low-amplitude vibration, and the bottom plate detected the oscillation amplitude of the micro(bio)sensor tip through a lock-in amplifier that operated in a frequency range of 200–500 kHz. The

characteristic resonance frequency of a micro(bio)sensor was determined by comparison of the frequency spectra in air and within the buffer solution (Fig. 2b). A line-scan approach curve was performed until a stop criterion of 5% change in the shear force magnitude was reached (Fig. 2c). The micro(bio)sensor was subsequently retracted to a distance of $25 \pm 0.2\,\mu m$ above the poplar wood slice for real-time measurement of $H_2O_2$ or glucose.

The circle sample holder with a glass bottom was used to fix a poplar wood slice (∅: -5 mm), which is compressed by the inlaid Teflon ring (∅: -3 mm) (Supplementary Fig. 1e). The Teflon ring also serve as an electrochemical cell for all measurements during the enzymatic degradation of wood cell walls. All measurements were conducted in 80 μL of either 50 mM sodium acetate buffer at pH 5.5, or 50 mM sodium phosphate buffer at pH 6.0, containing 50 mM KCl. The humidity inside the Faraday cage was kept at 70–90% to reduce electrolyte evaporation (Supplementary Fig. 1a). Five μL of deionized water was added after every hour to keep the total volume constant.

### H$_2$O$_2$-driven LPMO9C oxidation of wood slices

Nine poplar wood slices (thickness: ~25 μm, diameter: ~6 mm) were incubated with 100 μM ascorbic acid and 0.5 μM *Nc*LPMO9C in 1.5 ml potassium phosphate buffer (50 mM, pH 6.0). The mixture was kept in a circular shaking motion at 20 rpm at room temperature for at least 10 hours. Aqueous solutions of $H_2O_2$ were prepared from 30% hydrogen peroxide at the appropriate concentrations to give a molar feed rate of 10 μmol per hour. At the end, the wood slices were taken out and soaked in 1 M NaCl solution for one hour to release all the bound enzymes. Slices were rinsed thoroughly with deionized water prior to experiments.

### Enzyme labeling

Purified CDHIIA and LPMO9C from *N. crassa* were labeled with DyLight D550 and DyLight D633 *N*-hydroxy-succinimide (NHS) ester reagents, respectively (Thermo Fisher Scientific). The NHS ester reagents target any exposed α-amine of the N-terminus as well as ε-amines of lysine residues. 2.5 mg of the enzymes were dissolved in 1 mL of 100 mM sodium phosphate-buffered saline, pH 7.4. The enzyme solutions were mixed with DyLight reagents and vortexed to ensure complete mixing. The tubes were covered with aluminum foil and incubated at room temperature (22 °C) for 1 h. The unlabeled dye in the reaction mixture was separated on 250 μL of resin. The molar ratio of DyLight to protein for *Nc*CDHIIA was found to be 7.2 moles of dye per mole of protein and 5.72 moles of dye per mole of *Nc*LPMO9C. The purified and labeled *Nc*CDHIIA and *Nc*LPMO9C were aliquoted and stored at −20 °C in a light-proof environment until further use.

### Confocal laser scanning microscope set-up

The transverse poplar wood section was incubated with 1 μM of *Nc*CDHIIA and *Nc*LPMO9C. First, 5 μl of *Nc*LPMO were applied to the section and 3 min later, 5 μL of *Nc*CDHIIA. The enzyme mixture was incubated in the dark and at room temperature for 20 min. Then, the wood section was carefully washed with DI water to remove the excess of unbound enzymes and mounted on a glass slide. A coverslip was placed on top of the specimen and sealed with nail polish to avoid water evaporation. Sections were observed on an inverted confocal laser scanning microscope SP8 (Leica, Germany) with a ×100 1.4 NA oil immersion objective and a zoom factor of 2. The pixel size was set to 0.027 μm with a scan speed of 400 Hz. To minimize possible interference between channels, the images were taken in sequential mode. First, *Nc*LPMO9C labeled with DyLight D633 was imaged with the excitation wavelength of 636 nm and laser intensity of 10 and 100% gain. The emission was collected by a HyD detector with no gaining between 660 and 700 nm. *Nc*CDHIIA labeled with DyLight D550 was observed using an excitation wavelength of 516 nm with a laser intensity of 10%, and the emitted signal was collected by a HyD

detector over the range at 560–600 nm with 100% gain. Poplar lignin autofluorescence was excited at 405 nm with a laser intensity of 10%, and the emission was collected by a HyD detector between 415 and 550 nm with 100% gain. In all channels, a line average of 8 was used without frame accumulation. All confocal images were further deconvolved with Huygens Professional v.19.04 (Scientific Volume Imaging). The point spread function was theoretically calculated by LAS X software (Leica, Germany), as well as the background for all channels. The deconvolution was performed with the CMLE algorithm and the signal-to-noise ratio was set to 7 for LPMO and CDH channels and 10 for the lignin channel. The number of iterations was set to 40 with a stop criterion of 0.05.

## Data availability
Source data are available for Figs. 1–6 and Supplementary Figs. 2–8. Source data are provided with this paper.

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

## Acknowledgements

We thank Wolfgang Schuhmann for valuable discussions and support in setting up the SECM and Monika Debreczeny for help with the confocal laser scanning microscope. We thank Christopher Schulz for providing the material and protocol for the polyurethane coating. This work was supported by European Research Council through European Union's Horizon 2020 research and innovation program (ERC Consolidator Grant OXIDISE) under grant agreement no. 726396.

## Author contributions

H.C. and R.L. designed the study. H.C., N.G.A., A.B., and F.C. performed the experiments. H.C., D.K., F.C., L.S., N.G.A., S.S., and R.L. interpreted data. H.C. and R.L. wrote the initial manuscript. All authors contributed in revising and writing the final version of the manuscript.

## Competing interests
The authors declare no competing interests.
