## [Peer Review File · Nature Communications]

Investigating Lytic Polysaccharide Monooxygenase-assisted wood cell wall degradation with microsensorsReviewer #1 (Remarks to the Author):

=> It is not clear in SEM images the morphological difference between modified PB and the bare Pt electrodes. Authors should provide SEM images in higher magnification.

=> "The H₂O₂ microsensors were characterized by a pair of redox waves (Supplementary Fig. 1)." This graph is presented in Fig. S2.

=> "...is a necessary feature to perform measurements under natural, air-saturated conditions (Supplementary Fig. 2)." This graph is presented in Fig. S3.

=> "The high selectivity of the electrocatalyst for H₂O₂ reduction is consistent with previous reports and ensures an extremely low

background signal allowing the selective and sensitive detection of H₂O₂."

The selectivity of the sensor can not be attested just measuring the behavior in air-saturated conditions, since there are many other species presented in a cell, which could be electrochemically active.

=> "The amperometric response at 0.0 V vs Ag|AgCl needs time..."

Why the potential at 0.0 V was selected? Authors should present cyclic voltammograms before and after the presence of H₂O₂ to justify the use of 0.0 V.

=> Authors should provide additional analytical parameters, such as correlation coefficient, LOD, LOQ, Intra-day RSD, Inter-day RSD.

=> "The developed microsensor approach is currently the only available method to measure the peroxidase activity of

LPMO on an intact plant cell wall sample." Authors should highlight this information in the Introduction.

Reviewer #2 (Remarks to the Author):

The present manuscript written by Chang and coworkers demonstrated localized and time-resolved determination of Lytic Polysaccharide MonoOxygenase (LPMO) using piezo-controlled H₂O₂ microsensor. Recently, the contribution of hydrogen peroxide to the reaction of LPMO during biomass decomposition has been intensively discussed, but no conclusion has been reached as to whether the reaction occurs only in artificial systems or whether the microbes that use LPMO actually utilize such systems. This study is a very important paper that puts an end to such so-called "round-about discussions," and we believe that it is quite worthy of publication in Nature Communications. Although minor revisions are necessary, as described later, the reviewer strongly recommends acceptance of this paper.

Main Discussion:

Given the reactivity of CDH, it is rather surprising that previous papers have failed to detect the hydrogen peroxide produced by CDH. This result itself provides very important information on the decomposition process of plant cell walls. On the other hand, the reactivity of CDH to oxygen and the stability of reactive oxygen species are very much affected by compounds such as organic acids present in the vicinity and the local pH, and I believe that a discussion of these issues is essential. The reviewer encourage the authors to consider including such a discussion.

Another point of concern was the resolution of this experimental system. There is no need to add this as an experiment, but for the localization of LPMO and CDH in Figure 5, for example, how wide a range of hydrogen peroxide can be detected by this microsensor-based method? Scanning is considered to inevitably cover a wide area, but can the resolution be improved by reducing the size of the electrode or other measures in the future?

Minor point

The reviewing process is very inefficient without line number or page number. Please add for next review, if it is necessary.

In material and methods section, "H₂O₂-driven LPMO9C oxidation..." thickness and diameter include characters which cannot see in my PC environment.

Reviewer #3 (Remarks to the Author):

This research done by Chang et al. developed a microsensor that can be used to detect the "in situ" generation and consumption of H₂O₂ in plant samples. Especially, the authors using this design demonstrated the changes of H₂O₂ during LPMO degrades real wood samples. This development of the microsensor that enables the detection of real-time H₂O₂ is significant to the research of wood decomposition involving instantaneous fungal activities that are usually difficult to catch up by using routine bulk methods.

In general, this research was well designed, performed, and analyzed for data. The paper was well written and easy to follow. I believe it will have a broad audience if it is published in NC. In terms of the paper, I have several minor questions:

1. If the high concentration of H₂O₂ will damage the CDH/GOX while the enzymes produce H₂O₂? From Figure 3A, it seems this is not the case, how do the authors explain the H₂O₂ generation is not reactivating their corresponding enzymes?
2. In Figures 3 b and c, the controls without adding LPMO should be included.

Point-by-point response

Reviewer #1 (Remarks to the Author):

Q1: It is not clear in SEM images the morphological difference between modified PB and the bare Pt electrodes. Authors should provide SEM images in higher magnification.

Authors' response:

We have obtained a higher-magnification SEM image showing the Prussian blue electrocatalyst on the modified ultramicroelectrode. The 8000 × magnified image is now given in Figure 2C. The Prussian blue covers the platinum wire at the center of the ultramicroelectrode by forming a granule, which is shown as a close-up in the inset of Figure 2C. In Figure 2B the bare conductive platinum wire before the modification with Prussian blue is shown. The Pt electrode is seen as a bright white spot in the SEM image, because of the high number of generated secondary electrons. The structure of the bare platinum wire will – similar to the surrounding glass – also have scratches from the polishing material. However, for this study the diameter of the Pt electrode and its centric position on the electrode is of importance and not its surface roughness. After modification (Figure 1C) the platinum electrode is covered by Prussian blue and fewer secondary electrons are emitted, which demonstrates the successful modification. The difference between the unmodified and the Prussian blue-modified electrode is now better visible.

Q2: The H₂O₂ microsensors were characterized by a pair of redox waves (Supplementary Fig. 1). This graph is presented in Fig. S2.

Authors response:

Thank you for pointing our error out. The numbering has been corrected.

Q3: => ..."is a necessary feature to perform measurements under natural, air-saturated conditions (Supplementary Fig. 2)." This graph is presented in Fig. S3.

Authors response:

Thank you for pointing our error out. The numbering has been corrected.

Q4: The high selectivity of the electrocatalyst for H₂O₂ reduction is consistent with previous reports⁴⁰ and ensures an extremely low background signal allowing the selective and sensitive detection of H₂O₂." The selectivity of the sensor cannot be attested just measuring the behavior in air-saturated conditions, since there are many other species presented in a cell, which could be electrochemical active.

Authors response:

We measured under air-saturating conditions because that is the default condition in all experiments of this study. The reviewer is correct to point out that we should additionally study the effect of oxygen and other potential electroactive species on the calibration curve, which we did. To investigate the influence of oxygen we compared the original measurement to one in which we sparged the buffer with argon (anaerobic conditions). These data are now shown in Supplementary Figure 5a. No significant deviation of the calibration curve for hydrogen peroxide was observed between air saturated and anaerobic conditions.

To determine the influence of potential electroactive species on the detection of hydrogen peroxide under aerobic and anaerobic conditions, we prepared a poplar wood extract in the measurement buffer. Here we observed a small difference in the hydrogen peroxide sensitivity in presence of oxygen ($0.158 \mu\text{A } \mu\text{M}^{-1}$) and in the absence of oxygen ($0.194 \mu\text{A } \mu\text{M}^{-1}$). However, in this experiment a very concentrated extract from poplar wood (10 g in 100 mL buffer) was used and the concentration of electroactive species is assumed to be >100 times higher compared to the experiments using an immersed microtome section of poplar wood (<1 mg/mL). We want to point out that in all experiments performed with wood slices an *in situ* calibration was performed after the poplar wood slice was already submersed in the buffer. By this calibration method, the effect of electroactive species on the electrode is taken in account and we believe this to be the most accurate method to calibrate the microsensor.

What we failed to express in the original sentence was, however, that with this microelectrode hydrogen peroxide can be selectively determined in the presence of oxygen. We therefore modified the sentence to: "The high selectivity of the electrocatalyst for H_2O_2 reduction is consistent with previous reports⁴⁰ and ensures an extremely low background signal allowing the selective and sensitive detection of H_2O_2 in presence of O_2 ."

Q5: "The amperometric response at 0.0 V vs Ag|AgCl needs time..." Why the potential at 0.0 V was selected? Authors should present cyclic voltammograms before and after the presence of H_2O_2 to justify the use of 0.0 V.

Authors response:

The cyclic voltammograms of the H_2O_2 microsensor in absence and in presence of 50 or 100 μM H_2O_2 are given in Figure S 3. The results show that the current at a potential of 0.0 V vs Ag|AgCl increases with H_2O_2 concentration. The selection of 0.0 V vs Ag|AgCl as measurement potential is based on the observation that at this potential, a high response current with a low baseline noise was obtained. At this potential the distance to the midpoint redox potential of Prussian blue (0.2 V) is also far enough not to change the redox state of the catalyst. Finally, to exclude any possible influence of oxygen reduction (which we do not observe in the CV), we did not select a lower potential that is potentially more prone to oxygen interference.

Q6: Authors should provide additional analytical parameters, such as correlation coefficient, LOD, LOQ, Intra-day RSD, Inter-day RSD.

Authors response:

Additional analytical parameters of the H_2O_2 microsensor and the glucose microbiosensor are now provided in Supplementary Table 1 and 2. All the data were measured by using freshly prepared and calibrated H_2O_2 microsensors or glucose microbiosensors. Since the micro(bio)sensors were only used for one experiment and calibrated before and checked after each experiment no intra-day and inter-day RSD values were determined.

Q7: "The developed microsensor approach is currently the only available method to measure the peroxygenase activity of LPMO on an intact plant cell wall sample." Authors should highlight this information in the Introduction.

Authors response:

In order to emphasize that the microsensor setup represents the first method to study LPMO activity and its hydrogen peroxide consumption *in situ* using a natural substrate we modified the introduction:

Line 86: **Although LPMO activity can be measured by different assays, no currently available method is able to detect its activity at the microscale when bound to a natural substrate.** To investigate LPMO's peroxygenase activity on the surface of wood cell walls in a physiologically relevant environment, a H₂O₂ microsensor was **developed and** positioned closely above a poplar wood microtome slice by using a scanning electrochemical microscopy (SECM) platform. Since **a** microsensor is barely limited by mass transfer and only marginally depletes the analyte, the setup allows the real-time detection of H₂O₂ consumption by LPMO bound onto wood cell walls **for the first time.**

Reviewer #2 (Remarks to the Author):

The present manuscript written by Chang and coworkers demonstrated localized and time-resolved determination of Lytic Polysaccharide Mono Oxygenase (LPMO) using piezo-controlled H₂O₂ microsensor. Recently, the contribution of hydrogen peroxide to the reaction of LPMO during biomass decomposition has been intensively discussed, but no conclusion has been reached as to whether the reaction occurs only in artificial systems or whether the microbes that use LPMO actually utilize such systems. This study is a very important paper that puts an end to such so-called "round-about discussions," and we believe that it is quite worthy of publication in Nature Communications. Although minor revisions are necessary, as described later, the reviewer strongly recommends acceptance of this paper.

Main Discussion:

Q1: Given the reactivity of CDH, it is rather surprising that previous papers have failed to detect the hydrogen peroxide produced by CDH. This result itself provides very important information on the decomposition process of plant cell walls. On the other hand, the reactivity of CDH to oxygen and the stability of reactive oxygen species are very much affected by compounds such as organic acids present in the vicinity and the local pH, and I believe that a discussion of these issues is essential. The reviewer encourages the authors to consider including such a discussion.

Authors response:

Organic acids as well as other compounds produced by fungi are well known to be crucial for fungal biomass degradation (e.g. [1]). We believe that the conditions found *in situ* and local concentrations of oxygen, inhibitors, activators as well as local pH gradients play a major role to orchestrate enzyme activity. Those local conditions do affect the enzymes in this environment. In another study, we could show that LPMO is affected by different organic acids and interestingly strongly inhibited by oxalic acid which is commonly secreted by wood degrading fungi [2]. We have added a short explanation to the Discussion section:

“The production of H₂O₂ by CDH under natural conditions is regulated by many factors, such as the availability of phenol derived electron acceptors (which reduce the oxygen turnover), the pH which has a strong effect not only on the catalytic activity, but also the electron transfer to LPMO via CDH’s cytochrome domain. The presence of organic acids is an important factor that can therefore not only modulate the production and stability of H₂O₂, but has also shown to have a modulating effect on LPMO activity.”

Q2: Another point of concern was the resolution of this experimental system. There is no need to add this as an experiment, but for the localization of LPMO and CDH in Figure 5, for example, how wide a range of hydrogen peroxide can be detected by this microsensor-based method? Scanning is considered to inevitably cover a wide area, but can the resolution be improved by reducing the size of the electrode or other measures in the future?

Authors response:

The linear range of the hydrogen peroxide microsensors is typically between 20 μM to 200 μM and is sufficient for the performed measurements. The measurement of lower concentrations depends, as the reviewer points out, on smaller electrode diameters. The tip radius of the working electrode used within an SECM setup is crucial for its resolution and for the localized detection of analytes. In general, a higher resolution can be obtained using smaller electrode diameters, as has been shown in previous reports [3,4]. In our experiments an improved resolution would be possible using a smaller electrode. The reduction of the electrode diameter is currently ongoing, but the production of electrodes with a smaller diameter is difficult and less reproducible. So, currently the limiting factor is our equipment and the electrode production protocol. Information on the analytical parameters of the electrodes is available in the Supplemental Tables 1 and 2.

Topographical scanning of the wood sample was tried, but the consumption of hydrogen peroxide by the electrode turned out to be limiting this approach. Under scanning conditions, the electrode depletes the hydrogen peroxide in vicinity of the sample. Therefore, we measured at a fixed distance (the distance between the sample and the tip end of a micro(bio)sensor was adjusted to 25 μm using the shear-force based SECM), because a closer distance would lead to a very fast consumption of hydrogen peroxide and interfere with the measurement.

Minor point

Q3: The reviewing process is very inefficient without line number or page number. Please add for next review, if it is necessary.

Authors response:

We are sorry and added line numbers as well as page numbers for the revised version of the manuscript.

Q4: In material and methods section, "H2O2-driven LPMO9C oxidation..." thickness and diameter include characters which cannot see in my PC environment.

Authors response:

We did unfortunately not observe the bad PDF conversion of this section of the manuscript. In our version the presentation seemed to be correct. We suppose that the use of the "~" symbol potentially caused the problem. The missing part (Line 628ff.) is given here, but we hope that in the corrected version this will not occur again:

"Nine poplar wood slices (thickness: $\sim 25 \mu\text{m}$, diameter: $\sim 6 \text{ mm}$) were incubated with 100 μM ascorbic acid and 0.5 μM NcLPMO9C in 1.5 ml potassium phosphate buffer (50 mM, pH 6.0). The mixture was kept in a circular shaking motion at 20 rpm at room temperature for at least 10 hours. Aqueous solutions of H_2O_2 were prepared from 30% hydrogen peroxide at the appropriate concentrations to give a molar feed rate of 10 μmol per hour. At the end, the wood slices were taken out and soaked in 1 M NaCl solution for one hour to release all the bound enzymes. Slices were rinsed thoroughly with deionized water prior to experiments."

Reviewer #3 (Remarks to the Author):

This research done by Chang et al. developed a microsensor that can be used to detect the "in situ" generation and consumption of H₂O₂ in plant samples. Especially, the authors using this design demonstrated the changes of H₂O₂ during LPMO degrades real wood samples. This development of the microsensor that enables the detection of real-time H₂O₂ is significant to the research of wood decomposition involving instantaneous fungal activities that are usually difficult to catch up by using routine bulk methods.

In general, this research was well designed, performed, and analyzed for data. The paper was well written and easy to follow. I believe it will have a broad audience if it is published in NC. In terms of the paper, I have several minor questions:

Q1: If the high concentration of H₂O₂ will damage the CDH/GOX while the enzymes produce H₂O₂? From Figure 3A, it seems this is not the case, how do the authors explain the H₂O₂ generation is not reactivating their corresponding enzymes?

Authors response:

As stated by the reviewer, Figure 3A shows that the production of hydrogen peroxide by CDH or GOX is stable over 120 min, maybe levelling off a little bit in at later times (120-180 min) as seen in Figure 3B. The concentration of dissolved oxygen is also a little bit increasing during this time in Figure 3B, which indicates a small, but observable effect of the accumulated hydrogen peroxide concentration (130–200 μM) on the enzymatic activity of the hydrogen peroxide producing enzymes. In regard to the second part of the comment we interpret it as why in Figure 3C the hydrogen peroxide concentration after the addition of LPMO does not reach the same slope as before. The answer is, that we stopped the experiment after determining the hydrogen peroxide consumption rate by LPMO for which the experiment was designed. We did not wait until the LPMO was fully inactivated since we were not investigating LPMO inactivation. In Figure 3C an increase of the hydrogen peroxide concentration is seen after 210–230 min but not to the same extent, because part of the LPMO is still actively consuming its cosubstrate. This effect is most prominent with the lowest (0.5 μM) LPMO addition and not observed in the experimental time with the highest (3 μM) LPMO addition. Inactivation of LPMO (tested in Figure 4a), especially in presence of high hydrogen peroxide concentrations is known from previous studies and suggested to be responsible for non-linear progress curves during substrate conversion experiments [5]. The inactivation of LPMO by the addition higher amounts of hydrogen peroxide has been shown for cellulose-active LPMOs as well as for LPMOs active on chitin [6,7]. In order to clarify our interpretation, that high concentrations of H₂O₂ lead to the inactivation of LPMO and no significant inactivation of the hydrogen peroxide producing enzymes (GOx, CDH) was observed in our study we adapted the manuscript as follows (bold text: changes):

Line 425 f.: "The H₂O₂-driven LPMO catalysis **was observed for more than 2 h** in a near-natural environment, but eventually, the **high amount of H₂O₂ produced by the auxiliary enzymes** leads to LPMO inactivation. **The H₂O₂ producing enzymes are more stable towards their product than LPMO and are active even after accumulating H₂O₂ for up to 4 h in the experiments.**"

Q2: In Figures 3 b and c, the controls without adding LPMO should be included.

Authors response:

We have not included the data of the control experiment (no LPMO addition) in Figures 3B and 3C, because it shows the progression of the hydrogen peroxide production and would distract the reader from the key message that LPMO is consuming the hydrogen peroxide. The production and accumulation of hydrogen peroxide would continue to progress as is seen in Figure 4A, which shows the control experiment (in absence of LPMO). We feel that it fits better here, but have changed the figure caption to point out the control experiment (bold text).

Fig. 3 Detection of local H₂O₂ formation and consumption in the vicinity of wood cell walls. a The graph shows time courses of H₂O₂ formation by 2 mg mL⁻¹ cellobiohydrolases and 1 μM 1 μM *C. hotsonii* CDH variant (filled circles) or 2 mg mL⁻¹ cellobiohydrolases and 1 mg mL⁻¹ β-glucosidase in combination with 1 μM GOX (filled triangles) during hydrolysis of poplar wood cell walls **in the absence of LPMO**. In the control experiment (empty squares), only 2 mg mL⁻¹ cellobiohydrolase **has been** applied. **b** The change of O₂ and H₂O₂ concentration in the vicinity of wood cell walls before and after LPMO catalysis. The additions of 1 μM LPMO9C and 1.0 μM *N. crassa* CDHIIA are indicated by an arrow. **c** Time courses of H₂O₂ consumption by 0.5 μM, 1.0 μM, and 3.0 μM LPMO9C in combination with 1.0 μM *N. crassa* CDH. Their additions after 180 min are indicated by a gray arrow. In the experiments shown in panels **(b)** and **(c)**, 2 mg mL⁻¹ cellobiohydrolases and 1 μM *C. hotsonii* CDH variant are applied since the beginning to produce H₂O₂. All experiments were conducted in a 50 mM sodium acetate buffer, pH 5.5 at ~20 °C. **The reference experiment without addition of LPMO is given in Figure 4a (black circles)**. Data in panel **(a)** are shown as means values, and error bars show SD (n = 3, independent experiments). Data in panel **(c)** are shown as means values, and error bars show SD (n = 2, independent experiments).

References

1. Arantes, V.; Jellison, J.; Goodell, B. Peculiarities of Brown-Rot Fungi and Biochemical Fenton Reaction with Regard to Their Potential as a Model for Bioprocessing Biomass. *Appl. Microbiol. Biotechnol.* **2012**, *94*, 323–338, doi:10.1007/s00253-012-3954-y.
2. Breslmayr, E.; Poliak, P.; Požgajč, A.; Ludwig, R. Inhibition of the Peroxygenase Lytic Polysaccharide Monooxygenase by Carboxylic Acids and Amino Acids. **2022**.
3. Kwak, J.; Bard, A.J. Scanning Electrochemical Microscopy. Theory of the Feedback Mode. *Anal. Chem.* **1989**, *61*, 1221–1227, doi:10.1021/ac00186a009.
4. Xiong, Q.; Wu, T.; Song, R.; Zhang, F.; He, P. Theoretical and Experimental Verification of Imaging Resolution Factors in Scanning Electrochemical Microscopy. *Anal. Methods* **2021**, *13*, 1238–1246, doi:10.1039/d1ay00025j.
5. Eijsink, V.G.H.; Petrovic, D.; Forsberg, Z.; Mekasha, S.; Røhr, Å.K.; Várnai, A.; Bissaro, B.; Vaaje-Kolstad, G. On the Functional Characterization of Lytic Polysaccharide Monooxygenases (LPMOs). *Biotechnol. Biofuels* **2019**, *12*, 1–16, doi:10.1186/s13068-019-1392-0.
6. Bissaro, B.; Røhr, Å.K.; Müller, G.; Chylenski, P.; Skaugen, M.; Forsberg, Z.; Horn, S.J.; Vaaje-Kolstad, G.; Eijsink, V.G.H.H.; Forsberg, Z.; et al. Oxidative Cleavage of Polysaccharides by Monocopper Enzymes Depends on H₂O₂. *Nat. Chem. Biol.* **2017**, *13*, 1123–1128, doi:10.1038/nchembio.2470.
7. Kuusk, S.; Bissaro, B.; Kuusk, P.; Forsberg, Z.; Eijsink, V.G.H.; Sørli, M.; Valjamae, P. Kinetics of H₂O₂-Driven Degradation of Chitin by a Bacterial Lytic Polysaccharide Monooxygenase. *J. Biol. Chem.* **2018**, *293*, 523–531, doi:10.1074/jbc.M117.817593.

Reviewer #1

In comments to the editor the reviewer found all the issues to have been addressed

Reviewer #2 (Remarks to the Author):

Now the manuscript is suitably revised and I would like to suggest acceptance of the manuscript without revision.

Reviewer #3 (Remarks to the Author):

My concerns have been well considered in the revised version!